# Zinc and Boron Soil Applications Affect *Athelia rolfsii* Stress Response in Sugar Beet (*Beta vulgaris* L.) Plants

**DOI:** 10.3390/plants12193509

**Published:** 2023-10-09

**Authors:** Tamalika Bhadra, Chandan Kumar Mahapatra, Md. Hosenuzzaman, Dipali Rani Gupta, Abeer Hashem, Graciela Dolores Avila-Quezada, Elsayed Fathi Abd_Allah, Md. Anamul Hoque, Swapan Kumar Paul

**Affiliations:** 1Department of Agronomy, Bangladesh Agricultural University, Mymensingh 2202, Bangladesh; tamalikabhadra5@gmail.com (T.B.); mahapatra1977.cm@gmail.com (C.K.M.); 2Department of Soil Science, Bangladesh Agricultural University, Mymensingh 2202, Bangladesh; hosen.ss@bau.edu.bd (M.H.); anamul71@bau.edu.bd (M.A.H.); 3Institute of Biotechnology and Genetic Engineering, Bangabandhu Sheikh Mujibur Rahman Agricultural University, Gazipur 1706, Bangladesh; drgupta80@bsmrau.edu.bd (D.R.G.); 4Botany and Microbiology Department, College of Science, King Saud University, P.O. Box 2460, Riyadh 11451, Saudi Arabia; habeer@ksu.edu.sa (A.H.); 5Facultad de Ciencias Agrotecnológicas, Universidad Autónoma de Chihuahua, Chihuahua 31350, Mexico; gdavila@uach.mx (G.D.A.-Q.); 6Plant Production Department, College of Food and Agricultural Sciences, King Saud University, P.O. Box 2460, Riyadh 11451, Saudi Arabia; eabdallah@ksu.edu.sa (E.F.A.)

**Keywords:** sugar beet, ROS, antioxidant defense, *Athelia rolfsii* stress

## Abstract

Generation of reactive oxygen species (ROS) constitutes an initial defense approach in plants during pathogen infection. Here, the effects of the two micronutrients, namely, zinc (Zn) and boron (B), on enzymatic and non-enzymatic antioxidant properties, as well as malondialdehyde (MDA) contents in leaves and roots challenged with *Athelia rolfsii*, which cause root rot disease, were investigated. The findings revealed that Zn and B application to the potting soil alleviated the adverse effect of *A. rolfsii* on sugar beet plants and increased the chlorophyll content in leaves. The increased enzymatic antioxidant activities such as catalase (CAT), peroxidase (POX), and ascorbate peroxidase (APX), and non-enzymatic antioxidants such as ascorbic acid (AsA) were observed in Zn applied plants compared to both uninoculated and inoculated control plants. A significant rise in CAT activity was noted in both leaves (335.1%) and roots (264.82%) due to the Zn_2_B_1.5_ + Ar treatment, in comparison to the inoculated control plants. On the other hand, B did not enhance the activity of any one of them except AsA. Meanwhile, *A. rolfsii* infection led to the increased accumulation of MDA content both in the leaves and roots of sugar beet plants. Interestingly, reduced MDA content was recorded in leaves and roots treated with both Zn and B. The results of this study demonstrate that both Zn and B played a vital role in *A. rofsii* tolerance in sugar beet, while Zn enhances antioxidant enzyme activities, B appeared to have a less pronounced effect on modulating the antioxidant system to alleviate the adverse effect of *A. rolfsii.*

## 1. Introduction

Sugar beet (*Beta vulgaris* L.) holds significant global importance as a cultivated sugar crop, ranking second only to sugarcane [1]. This crop accounts for nearly 30% of the world’s sugar production for human consumption [2,3]. Unlike sugarcane, sugar beet has a relatively short growth period, with its mature tap root containing sucrose (15–20%), water (75–76%), non-sugars (2.6%), and pulp (4–6%) [4,5]. Beyond sugar extraction, both the foliage and beet pulp find utility as cattle feed [6]. More recently, sugar beet has gained prominence as a crucial resource for ethanol production as a biofuel [7]. Nonetheless, the production of sugar beet faces significant challenges posed by various diseases. Among these, sugar beet root rot stands out as a destructive fungal ailment caused by *Athelia rolfsii* (Curzi) Tu and Kimbrough (anamorph, *Sclerotium rolfsii* Sacc.). This fungus primarily infects the crown area or regions near the stem close to the soil surface, resulting in the decay of infected tissues. Such infections can lead to substantial yield losses, often reaching up to 50% [8]. Initially, sugar beet leaves show yellowing and wilting of plants, then soft and water-soaked fleshy roots appear. The disease may lead to kill the plant due to severe infection [9]. This pathogen generates a variety of secondary metabolites that play crucial roles in attaching to plant surfaces and protecting sclerotia against adverse environmental conditions [10]. A recent study suggests that *A. rolfsii* significantly disrupts proteins and metabolites associated with both photosynthesis and antioxidant mechanisms within chloroplasts and mitochondria, impacting carbohydrate and amino acid metabolism processes [11].

Plants activate many defense mechanisms in reaction to biotic stress, as it may be linked either directly to responses to the immune system or to a signal transduction pathway. Production of ROS such as singlet oxygen (^1^O_2_), superoxide (O_2_^•−^), hydrogen peroxide (H_2_O_2_), and hydroxyl radical (OH^•^) is one of the defense mechanisms employed by plants against pathogens. However, an excess of ROS inside the plant cell can lead to cellular damage. To mitigate this risk, plant cells generate various types of antioxidants to reduce the level of ROS. This detoxification involves both enzymatic and non-enzymatic antioxidant constituents [12,13].

Superoxide dismutase (SOD), catalase (CAT), ascorbate peroxidase (APX), glutathione peroxidase (GPX), and peroxidase (POX) are enzymatic antioxidants, while ascorbic acid (AsA) and glutathione (GSH) are non-enzymatic antioxidants. These enzymes are present in nearly all subcellular units. Generally, an organelle contains numerous enzymes capable of scavenging ROS [13]. The ascorbate–glutathione cycle is the major hydrogen peroxide-detoxification system in plant chloroplasts, in which APX is a vital enzyme active at a very low level of H_2_O_2_ in the cell. The APX uses AsA as a precise electron donor to convert H_2_O_2_ to water. The ROS scavenging function of APX and ascorbate–glutathione cycle is not limited to chloroplasts but also extended to mitochondria, peroxisomes, and cytosol [14]. Similar to APX, CAT also exhibits a high affinity for H_2_O_2_ but remains active at relatively higher concentrations. In plant–pathogen interactions, POX also plays a vital role in supporting cell wall formation through the generation of lignin, dityrosine bonds, and additional cross-links. POX also acts as scavenger for H_2_O_2_ within cells by catalyzing the oxidation of numerous substrates using H_2_O_2_. The AsA functions as a co-factor for various enzymes and directly detoxifies ROS by neutralizing the superoxide singlets. Moreover, pathogen attacks can lead to membrane lipid peroxidation, resulting in an increase in malondialdehyde (MDA) content in cells, which is a secondary product produced during lipid peroxidation [15,16].

Zinc and boron are two micronutrients that have been reported to be essential for normal plant growth and for protecting plants against adverse environments [17]. Of them, Zn plays a key role in plant metabolism, functioning as an enzyme co-factor in various reactions such as metabolic reactions, energy transfer, and protein synthesis [18,19,20]. In addition, Zn increases the expression of stress proteins by interacting with plant hormones thereby stimulating the antioxidant enzymes to counteract various stresses such as salinity and drought [21,22]. Similarly, B has also been shown to alleviate the deleterious effects of ROS by activating antioxidant enzymes under abiotic stress [23,24]. Boron has also been involved in controlling plant disease by directly affecting cell membrane integrity, stability, and rigidity, resulting in increased resistance to cell permeability and metabolic activities [25]. Recent studies have suggested that both Zn and B inhibit pathogen growth and development by interfering with the signaling pathways involved in pathogenesis [18,26]. Furthermore, the judicious application of Zn and B can enhance the defensive system in plants against some fungal infections [18,20,25,26,27].

Recently, Paul et al. [8] assessed that 9.74% of plants in sugar beet fields at Agronomy Field Laboratory, BAU, Bangladesh, were infested with *A. rolfsii*, exhibiting root rot symptoms. Notably, the application of Zn and B in sugar beet plots partially suppressed root and crown rot symptoms in the sugar beet field (unpublished data). Despite sclerotium root rot posing a significant challenge in sugar beet cultivation worldwide, the potential for disease control through the utilization of Zn and B has not previously been investigated. A recent study conducted by Rerhou et al. [28] has shed light on the management of root rot disease in sugar beet by applying micronutrients, including Zn and B. Nevertheless, the precise mechanisms underlying how Zn and B effectively suppress *A. rolfsii* in sugar beet plants remain undisclosed. It is conceivable that these micronutrients might trigger the activation of the antioxidant system within sugar beet plants, assisting the plants in fortifying their defense mechanisms against *A. rolfsii*. Consequently, investigating the roles of Zn and B could present a promising avenue to diminish the incidence of root rot disease caused by *A. rolfsii*, thereby bolstering resistance, and promoting plant growth. The present study evaluated the contributions of Zn and B to the activation of the defense mechanisms in sugar beet plants against *A. rolfsii*.

## 2. Materials and Methods

### 2.1. Planting Material and Experimental Design

The experiment was conducted at the Agronomy Field Laboratory, Department of Agronomy, Bangladesh Agricultural University, Mymensingh, from November, 2019 to February, 2020. The tropical sugar beet variety PAC-60008 was used for the experiment. The study was laid out in a randomized complete block design (RCBD) and replicated three times. Similar size of plastic pots (depth 15 cm, top diameter 17 cm, opening surface area 0.023 m^2^) were used to conduct the study. The study consisted of the following treatments- (i) uninoculated control (Zn_2_B_0_), (ii) inoculated control (Zn_2_B_0_ + Ar; *Athelia rolfsii*), (iii) Zn_2_B_0_ (Zn 2.0 kg ha^−1^), (iv) Zn_2_B_0_ + Ar, (v) Zn_4_B_0_ (Zn 4.0 kg ha^−1^), (vi) Zn_4_B_0_ + Ar, (vii) Zn_0_B_1.5_ (B 1.5 kg ha^−1^), (viii) Zn_0_B_1.5_ + Ar, (ix) Zn_0_B_3_ (B 3 kg ha^−1^), (x) Zn_0_B_3_ + Ar, (xi) Zn_2_B_1.5_, (xii) Zn_2_B_1.5_ + Ar, (xiii) Zn_2_B_3_, (xiv) Zn_2_B_3_ + Ar, (xv) Zn_4_B_1.5_, (xvi)Z_4_B_1.5_ + Ar, (xvii) Z_4_B_3_, and (xviii) Z_4_B_3_ + Ar.

### 2.2. Pot Preparation, Fertilizer Application and Seed Sowing

To perform the study, soils were collected from the medium-high land of the Agronomy Field Laboratory at BAU. The soil possesses a silty loam texture with a pH 6.9, electrical conductivity (EC) 0.4 dS/m, organic carbon 1.00%, N 0.09%, P 1.60 ppm, K 0.10% meq/100g soil, Ca 8.30 meq/100 g soil, Mg 3.29 meq/100 g soil, S 2.98 ppm, Zn 0.21 ppm, and B 0.23 ppm. The soil was properly air dried, crushed with a wooden hammer, and thoroughly mixed. It was then cleaned and autoclaved. Finally, an equal amount of soil (10 kg) was used to fill each pot. Inorganic fertilizers were mixed thoroughly with the soil during pot filling [5]. Seeds were directly shown in the pots, with three seedlings in each pot, and kept in a growth room at a temperature of 20–25 °C and 12 h/12 h light/dark photoperiods. Light irrigation was applied every alternate day.

### 2.3. Fungi Culture, Inoculation and Sample Collection

The *A. rolfsii* isolate BTSB2 was used for inoculation assay [8]. Fungal inoculum was prepared using the technique reported by Figueredo et al. [29]. Briefly, wheat seeds were autoclaved and placed in a 50 mL flask. Approximately, 50 mm agar block from the actively growing *A. rolfsii* culture were cut and placed into the flask containing wheat seeds. The flask was then incubated for 10–15 days at 25 °C with a 12-h photoperiod. Infected wheat seeds were placed adjacent to the collar region of sugar beet plants 42 days after emergence and leaf and root samples were collected for analysis 7 days after the fungal inoculum application. The survival rate of the inoculated plants was calculated 20 days after the pathogen application and expressed as a percentage.

### 2.4. Determination of Soil-Plant-Analysis Development (SPAD) Value

A transportable SPAD meter (Model SPAD-502, Minolta Crop., Ramsey, NJ, USA) was used to determine the greenness (chlorophyll content) of sugar beet leaves 7 days after pathogen inoculation. Five fully expanded new leaves were selected SPAD values recording of beet plants. The average data were collected from the middle portion of the five leaves per plant between 9:00 to 10.00 a.m. [5].

### 2.5. Enzyme Extraction and Assay

Two plants from each replication (e.g., six plants per treatment) were taken and separated into shoots and roots. The collected leaf and root samples were then used for further analysis. Initially, 1 g of green leaves and roots was thoroughly blended and homogenized in a pre-chilled mortar-pestle, adding 5 mL of Tris–HCl buffer (50 mM, pH 8.0) for CAT and KH_2_PO_4_ buffer (50 mM, pH 7.0) for POX and APX analysis. The homogenized samples were collected and centrifuged at 5000 rpm for 20 min at 4 °C. Assay of antioxidant enzymes (CAT, POX, and APX) was conducted using the supernatant.

The CAT activity was determined following the technique described by Aebi [30]. The reaction mixtures comprised Tris–HCl buffer (50 mM, pH 8.0), EDTA (0.25 mM), H_2_O_2_ (20 mM), and 100 μL of enzyme extract (supernatant). The H_2_O_2_ was added to start the reaction and the CAT activity was measured at 240 nm wavelength.

Activity of POX was measured according to the method described by Nakano and Asada [31]. The reaction buffer solution included KH_2_PO_4_ buffer (50 mM, pH 7.0), EDTA (0.1 mM), H_2_O_2_ (0.1 mM) and guaiacol (10 mM). The reaction was initiated by adding 50 µL of sample solution to the reaction buffer (950 µL). The activity was determined at a wavelength of 470 nm for 30 sec, using an extinction coefficient 26.6 mM^−1^ cm^−1^.

For the determination of APX activity, the reaction mixture was prepared by adding KH_2_PO_4_ buffer (50 mM, pH 7.0), EDTA (0.1 mM), H_2_O_2_ (0.1 mM), and 0.5 mM ascorbate to the enzyme extract (50 µL). The activity was measured at 290 nm wavelength for 1 min, using an extinction coefficient 2.8 mM^−1^ cm^−1^ [31].

### 2.6. Estimation of the Ascorbic Acid (AsA) Content

The AsA content was calculated following the method described by Rangana [32]. The plant sample mixture, along with acetic acid (5 mL) and 3% of meta phosphate (5 mL) were titrated against 2,6 dichlorophenol indophenol until the pink color appeared. The concentration was determined in comparison with the standard solution.

### 2.7. Determination of the Malondialdehyde (MDA) Content

Oxidative damage in the leaf and root cells was estimated as the content of total 2-thiobarbituric acid (TBA)-responsive substances and expressed as equivalents of MDA following the method of Cakmak and Horst [33]. The amount of MDA was determined by measuring the absorbance at 532 nm and correcting for non-specific turbidity by subtracting the absorbance at 600 nm. An extinction coefficient of 155 mM^−1^ cm^−1^ was applied to calculate the MDA, expressed in µmol g^−1^ fresh weight [34].

### 2.8. Statistical Analysis

The statistical analyses were performed using the software package MSTAT-C, and mean differences were assessed using Duncan’s Multiple Range Test (DMRT) [35].

## 3. Results

### 3.1. Effect of Zn and B on Plant Survival under A. rolfsii Stress

All the plants survived in the uninoculated control (Zn_0_B_0_), although some yellowing of leaves was observed. On the other hand, *A. rolfsii* (inoculated control) treated plants showed wilting within 6–10 days after inoculation. Plants died within 20–25 days after inoculation, and infected roots became rotten and covered with mycelial mass (Figure 1B). However, all the plants treated with Zn and B and whether or not they were inoculated with *A. rolfsii*, survived. Nevertheless, the development of lesions in sugar beet roots due to the inoculation of *A. rolfsii* varied in various combinations of Zn and B treatments (Figure 1).

### 3.2. Effect of Zn and B on Chlorophyll Content (SPAD Value)

The application of Zn and B increased the chlorophyll content in sugar beet leaves. However, *A. rolfsii* treatment significantly reduced chlorophyll content, with the lowest (18.96) recorded in the inoculated control compared to the uninoculated control (25.03). Elevated chlorophyll content was recorded in plants treated with Zn_4_B_1.5_ (69.50) which was 177.67 and 266.56% higher in comparison to the uninoculated and inoculated control, respectively (Figure 2).

### 3.3. Effect of Zn and B on CAT Activity

The CAT activity in leaves and roots was significantly influenced by various treatments (Figure 3A). In the leaves of Zn treated plants, increased CAT activity was recorded in both *A. rolfsii* inoculated and uninoculated plants compared to the uninoculated control (Zn_0_B_0_). Interestingly, decreased CAT activity was detected in uninoculated B applied plants but increased in pathogen inoculated plants. The lowest CAT activity was recorded in plants treated only with pathogen (Zn_0_B_0_ + Ar), measuring 45.06 µmol min^−1^ g^−1^ FW. The highest CAT activity (196.34 µmol min^−1^ g^−1^ FW) was recorded for Zn_2_B_1.5_ + Ar which was 335.1% higher than inoculated control (Figure 3A).

In the root, a similar trend in CAT activity was observed. A significant increase in CAT activity (71.98 µmol min^−1^ g^−1^ FW) was noticed in the roots of Zn_4_B_0_ + Ar treated plants, which was 264.82% higher than the inoculated control plants, followed by Zn_2_B_0_ + Ar (68 µmol min^−1^ g^−1^ FW), Zn_2_B_1.5_ + Ar (61.75 µmol min^−1^ g^−1^ FW), Zn_2_B_3_ + Ar (58.84 µmol min^−1^ g^−1^ FW) and Zn_4_B_3_ + Ar (58.09 µmol min^−1^ g^−1^ FW) treated plants. Boron treatments did not enhance CAT activity in sugar beet roots. CAT exhibited a substantial decline in its activity in inoculated control (Zn_0_B_0_ + Ar) compared to the uninoculated plants (Figure 3B).

### 3.4. Effect of Zn and B on POX Activity

The sugar beet plants exhibited a similar pattern of POX activity as CAT in response to the application of Zn and B under *A. rolfsii* treatment. In both leaves and roots, higher POX activity was detected in plants treated with Zn compared to the uninoculated (Zn_0_B_0_) and inoculated (Zn_0_B_0_ + Ar) control, while decreased POX activity was detected in plants treated only B but it increased in pathogen inoculated plants. The highest POX activity (1.96 µmol min^−1^ g^−1^ FW) was observed in Zn_2_B_1.5_ + Ar treated plants, which was 43% higher than inoculated control (1.37 µmol min^−1^ g^−1^ FW) in leaves. The lowest POX activity was found in Zn_0_B_3_ (0.53 µmol min^−1^ g^−1^ FW) (Figure 4A).

In the roots, the highest POX activity (13.16 µmol min^−1^ g^−1^ FW and 12.23 µmol min^−1^ g^−1^ FW) was observed in Zn_2_B_0_ and Zn_2_B_1.5_ treated plants with pathogen inoculation, which was 26.05% and 17.14% higher than the inoculated control. The lowest POX activity was recorded in Zn_0_B_1.5_ (4.36 µmol min^−1^ g^−1^ FW) and Zn_0_B_3_ (3.81 µmol min^−1^ g^−1^ FW) treated plants (Figure 4B).

### 3.5. Effect of Zn and B on APX Activity

The APX activity in sugar beet leaves and roots is significantly affected by various treatments. Pathogen inoculation increased the APX activity in both Zn and B treated plants. However, a decreased APX activity was recorded in B applied plants compared to both inoculated and uninoculated control plants. In leaves, the highest APX activity (0.86 µmol min^−1^ g^−1^ FW) was recorded for Zn_2_B_0_ + Ar, which was 109.75% higher than inoculated control, whereas the lowest APX activity was recorded in Zn_0_B_1.5_ (0.37 µmol min^−1^ g^−1^ FW) (Figure 5A).

Similarly, in root samples, the APX activity also showed significant increase when treated with Zn. The B treated plants showed decreased APX activity, but it significantly increased in pathogen inoculated plants. The highest APX activity was recorded for Zn_2_B_1.5_ + Ar (13.27 µmol min^−1^ g^−1^ FW) treated plants, which was 131.58% higher than the inoculated control. The lowest APX activity was recorded in Zn_0_B_1.5_ (4.71 µmol min^−1^ g^−1^ FW), which was statistically similar to the Z_0_B_3_ treatment (Figure 5B).

### 3.6. Effect of Zn and B on AsA Concentration

The AsA levels of sugar beet leaves and roots showed significant variation among the treatments (Figure 6). An increase in AsA levels was noticed in Zn and B treated plants, both in leaves and roots, with a few exceptions. The highest AsA concentration (8.62% mg) was recorded in the leaves of *A. rolfsii* inoculated plant treated with Zn and B (Zn_4_B_3_ + Ar), which was 30.60% higher than the inoculated control (Z_0_B_0_). The AsA content in Zn_4_B_3_ + Ar was statistically identical to the treatments of Z_4_B_1.5_, Zn_4_B_1.5_ + Ar, Zn_2_B_3_, Zn_0_B_3_, and Zn_2_B_0_ in leaves (Figure 6A).

An increased AsA concentration was recorded in the roots of inoculated control compared to the uninoculated control (Zn_0_B_0_) in Zn treated plants. Interestingly, a decreased AsA concentration was observed in the roots that were only treated with B (Figure 6B). The highest AsA concentration (4.1% mg) was observed in Zn_4_B_0_ + Ar, which was on par with Zn_4_B_3_, while the lowest was in Zn_0_B_3_ (1.99% mg).

### 3.7. Effect of Zn and B on MDA Concentration

The increase in MDA is a sign of membrane impairment at the cellular level, and under this stress condition, lipid peroxidation is expressed [36]. The amount of MDA in sugar beet leaves and roots was significantly influenced by various treatments. The MDA content increased in plants inoculated with Ar alone compared to the uninoculated control but decreased in plants treated with Zn and B, with or without Ar (Figure 7). In leaves, the MDA concentration (0.0167 µmol g^−1^ FW) in inoculated control (Zn_0_B_0_ +Ar) was 35.77% higher than the uninoculated control (0.0123 µmol g^−1^ FW). Similarly, in roots, a higher MDA concentration (0.137 µmol g^−1^ FW) was recorded in inoculated control (Zn_0_B_0_ + Ar), while a lower amount was found in Zn_2_B_3_ (0.0036 µmol g^−1^ FW) treated plants. However, plants treated with Zn and B significantly decreased the MDA contents both in leaves and roots, even after the inoculation of the pathogen (Figure 7A,B).

## 4. Discussion

Antioxidative systems in plants generally provide a defense mechanism against the harmful effects of ROS generated during interaction with pathogens [37,38,39,40]. Antioxidant enzymes such as CAT, APX, and POX play important roles in detoxifying ROS. Several studies have suggested that micronutrients suppress pathogen growth either by direct inhibition or by modulating the antioxidant protection mechanism [41,42,43]. In this paper, we demonstrated the differential changes in antioxidant enzymes in sugar beet plants resulting from the application of Zn and B during *A. rolfsii* treatment. The results obtained from this study support the idea that Zn and B participate in a protective activity in sugar beet plants against *A. rolfsii* by modulating antioxidant enzymes.

Micronutrients such as Zn and B are considered essential factors for normal plant growth and for defending plants against hostile environments [17,18,19,20,25,26]. In this study, we observed that *A. rolfsii* causes root rot symptoms in plants and leads to the death of entire plants (Figure 1); while Zn and B alleviate the adverse effects of *A. rolfsii* and enhance the growth of sugar beet plants. Moreover, leaf chlorophyll content was reduced by the inoculation of *A. rolfsii*, but it significantly increased in Zn and B treated plants (Figure 2). Pathogen stress reduces the photosynthetic rate in plants due to damage to the photosynthetic mechanism, which can interrupt the food producing system [27,29]. Zn and B treatment significantly increases the leaf chlorophyll content, enriching sugar beet growth due to enhanced photosynthetic activity under pathogen stress. These results are in agreement with Noman et al. [44] and Aydin et al. [23], who found that Zn and B application significantly increase chlorophyll content in radish and tomato plants. Numerous studies advocated that the appropriate dose of Zn and B improve growth and production in various crops, including rice, wheat, maize, and chickpea [45,46,47,48].

Upon pathogen recognition, plants initiate the production of elevated levels of ROS. These ROS can damage the cellular membrane lipids of the plants, resulting in subsequent membrane permeability [49]. Membrane lipids are important targets of ROS due to their unsaturated nature, which is involved in the accumulation of MDA; this factor is used to determine the degree of oxidative stress damage. In the study, the application of Zn and B apparently decreased the MDA content in plants, thus improving the comparative membrane permeability of sugar beet plants and enhancing their survival rate under pathogen challenges (Figure 1 and Figure 7). Several studies have reported the role of Zn and B in decreasing the MDA content under various stresses. Zn is an essential element for all metabolic processes, including photosynthesis. Tufail et al. [50] reported that Zn application in rice plants enhances photosynthesis, which might improve the integrity of the cell membrane. On the other hand, B acts as a signal molecule that modulates the expression of enzymes needed for cell wall synthesis and assembly [40,51]. These findings from this study can be correlated with the findings that Zn and B prevent the peroxidation of membrane constituents, thus stabilizing the cell membrane by altering the expression of cell wall synthesis genes or enhancing the metabolic process in the sugar beet plant.

We observed an elevation in CAT, APX, and POX activity in Zn treated plants compared to both uninoculated and inoculated plants. CAT activity was notably higher in Zn-treated plants both before and after inoculation of *A. rolfsii*. However, in the inoculated control group (*A. rolfsii* treated plant), CAT activity exhibited a decrease compared to the uninoculated control (Zn_0_B_0_). It has been suggested that CAT might suppress biotrophic pathogens while encouraging necrotrophic pathogens through host–pathogen interaction [39,52]. The reduction in CAT activity in inoculated control plants likely contributes to the facilitation of pathogen infection. Conversely, the elevated CAT activity observed in Zn-treated plants, in contrast to untreated plants, underscores their enhanced ability to detoxify H_2_O_2_. Other studies have also highlighted the role of CAT in combating various fungal pathogens in important crops such as lettuce, oilseed rape, non-heading Chinese cabbage, and sunflower [37,39,53,54,55].

The POX is a well-known pathogenesis-associated protein that helps cell wall lignification and suberization, creating a physical barrier against invading pathogens for localized control [34,39]. Treatment with Zn increased the POX activity in both inoculated and uninoculated plants. Similarly, APX activity and AsA content were also increased in Zn treated plants. Since Zn application significantly enhanced POD, APX, and AsA activity and reducing MDA content, the increased survival rate of sugar beet plants under *A. rolfsii* stress could be attributed to the increased CAT, POD, APX, and AsA activity, which mitigates the harmful effects of ROS generated during pathogen infection. These results are consistent with other studies that Zn amendments or supplementation enhance the activities of various defense enzymes in plants, triggering systemic resistance against pathogens [17,18,20,25,27]. Zn plays a crucial role as a structural element in zinc finger proteins, and several studies suggest that these proteins regulate the expression of key ROS-scavenging enzymes such as CAT, SOD, POD, and APX, thereby imparting resistance to stressors [56]. Consequently, it is plausible to hypothesize that Zn treatment in inoculated plants likely activates zinc finger proteins, which subsequently stimulate a cascade of defense enzymes, thus fortifying the plant’s defenses against *A. rolfsii*.

Boron is another crucial micronutrient for plant growth and development. Interestingly, an increase in CAT, POX, and APX activity was observed in plants grown in boron-deficient soil (uninoculated control plants) compared to soil amended with B. This observation aligns with findings by Song et al. [57] and Wang et al. [58] where they noted that either excessive B or B deficiency induces stress in plants, consequently, leading to heightened activity of antioxidant enzymes in sugar beet plants. Optimum B doses, however, appear to mitigate the oxidative stress brought about by B-related stress. Nevertheless, the inoculation of *A. rolfsii* only slightly enhanced the activity of CAT, POX, and APX in plants grown in B-amended soil, in comparison to both uninoculated and inoculated control plants. The results of the study suggest that B aids in maintaining ROS homeostasis under conditions of B-deficiency by regulating the antioxidant enzyme activity. However, it does not seem to suppress the growth of *A. rolfsii* through the enhancement of antioxidant defense mechanisms. Boron is an integral part of cell wall structure, and B-deficiency leads to a decrease in the mechanical strength of the cell wall by altering cellulose, pectin, and lignin biosynthesis [51]. Several studies have indicated that the application of optimum doses of B in plants enhances the integrity of cell membranes and cell wall, which a significant factor is contributing to increased resilience during pathogen infection [24,25]. It is possible that B might promote the stability and rigidity of the cell wall structure, thus inhibiting the pathogen invasion in sugar beet plants resulting in a higher survival rate. On the other hand, B has been reported as an antifungal agent that directly inhibits the growth of numerous pathogenic fungi [40,41]. Furthermore, recent research has also proposed that both Zn and B suppress fungal growth by enhancing the endophytic biocontrol bacterial community thereby enhancing the plant growth [59]. Hence, ensuring proper plant nutrition is one of the essential approaches for the sustainable management of plant disease.

## 5. Conclusions

This study is the first to describe the regulation of ROS in the sugar beet-*A. rolfsii* pathosystem through the application of Zn and B. It is evident that both Zn and B help to alleviate the adverse effects of *A. rolfsii* in sugar beet plants and increase chlorophyll content. Both Zn and B play distinct roles in modifying the antioxidant defense response within sugar beet plants when faced with *A. rolfsii* infection. Zn has shown the ability to enhance the activity of antioxidant enzymes, whereas B appeared to have less pronounced effects on modulating the antioxidant system to mitigate the detrimental impacts triggered by *A. rolfsii*. However, a more comprehensive study is needed to explore the antifungal effects of Zn and B on *A. rolfsii*, examine the regulation of other antioxidant enzymes, and investigate the modulation of beneficial microorganisms in soil by the application of Zn and B in sugar beet cultivation system. This, in turn, could help to mitigate the detrimental impacts of *A. rolfsii* on sugar beet plants.

## Figures and Tables

**Figure 1 plants-12-03509-f001:**
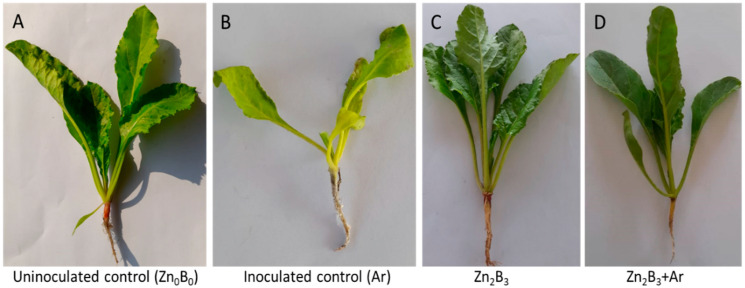
Effect of Zn and B on plant survival under *A. rolfsii* stress. (**A**) Uninoculated control (Zn_0_B_0_), (**B**) inoculated control (Ar), (**C**) Zn_2_B_3_; and (**D**) (Zn_2_B_3_ + Ar).

**Figure 2 plants-12-03509-f002:**
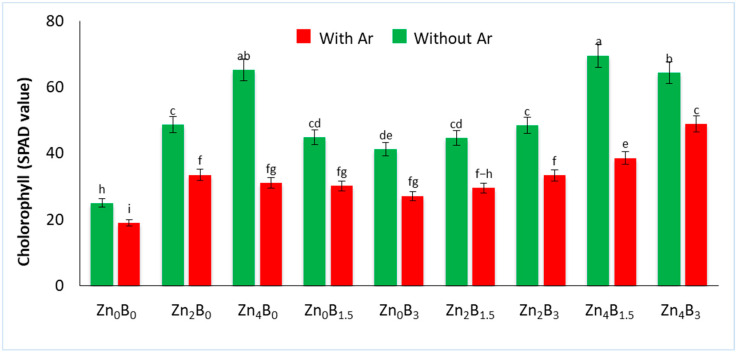
Effect of Zn and B on chlorophyll content in sugar beet plant under *A. rolfsii* stress. Data followed by the same letter are not significantly different by DMRT test at *p* < 0.05.

**Figure 3 plants-12-03509-f003:**
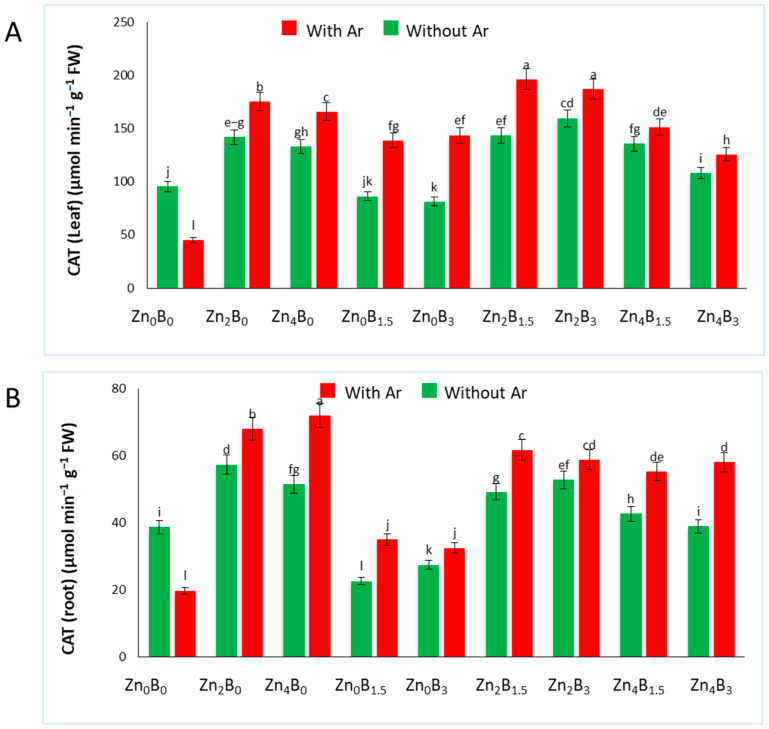
Catalase activity in (**A**) leaf and (**B**) roots of sugar beet plants. Plants were grown with/without Zn and B amended soil. Samples were collected 7 days after inoculation of *A. rolfsii*. Data are presented as treatment mean ± standard error. Data followed by the same letter are not significantly different by DMRT test at *p* < 0.05.

**Figure 4 plants-12-03509-f004:**
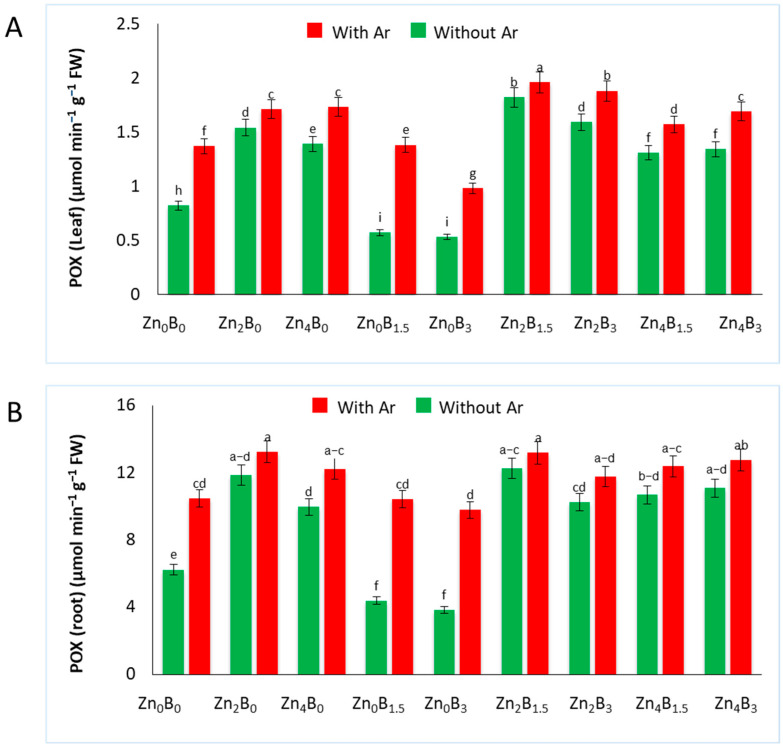
Peroxidase activity in (**A**) leaf and (**B**) roots of sugar beet plants. Plants were grown with/without Zn and B amended soil. Samples were collected 7 days after inoculation of *A. rolfsii*. Data are presented as treatment mean ± standard error. Data followed by the same letter are not significantly different by DMRT test at *p* < 0.05.

**Figure 5 plants-12-03509-f005:**
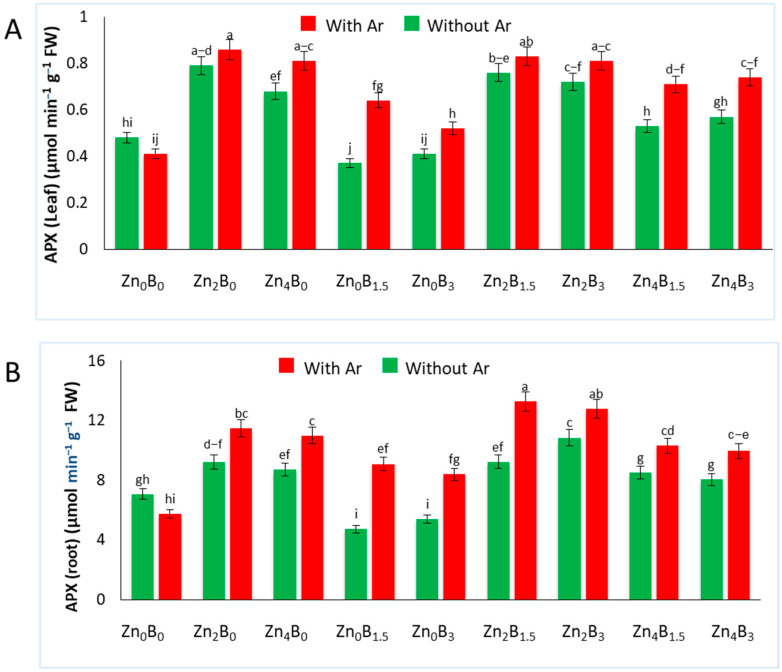
Ascorbate peroxidase activity in (**A**) leaf and (**B**) roots of sugar beet plants. Plants were grown with/without Zn and B amended soil. Samples were collected 7 days after inoculation of *A. rolfsii*. Data are presented as treatment mean ± standard error. Data followed by the same letter are not significantly different by DMRT test at *p* < 0.05.

**Figure 6 plants-12-03509-f006:**
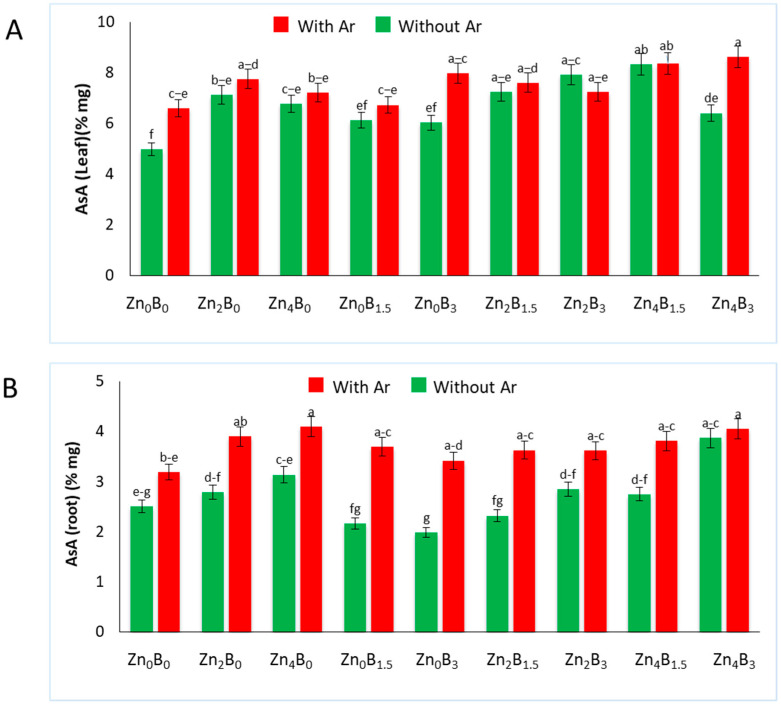
The content of ascorbic acid in (**A**) leaf and (**B**) roots of sugar beet plants. The plants were grown in soil amended with or without Zn and B. Samples were collected 7 days after inoculation of *A. rolfsii*. The data are presented as the means of the treatment ± standard error. Data labeled with the same letter are not significantly different based on the DMRT test at *p* < 0.05.

**Figure 7 plants-12-03509-f007:**
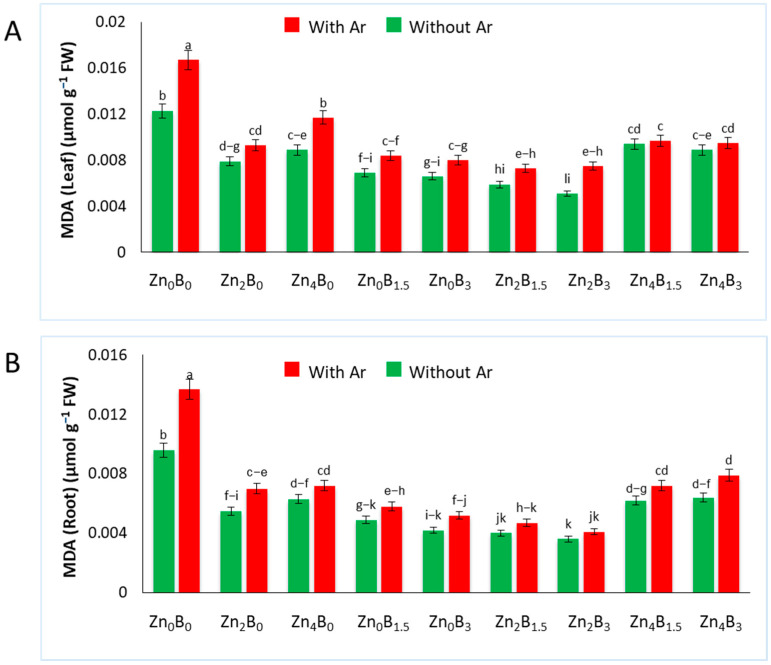
Malondialdehyde content in (**A**) leaf and (**B**) roots of sugar beet plants. The plants were grown in soil amended with or without Zn and B. Samples were collected 7 days after the inoculation of *A. rolfsii*. The data are presented as the means of the treatment ± standard error. Data sharing the same letter are not significantly different according to the DMRT test at *p* < 0.05.

## Data Availability

The data are contained within the article.

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
