# Peer review of "Zinc and Boron Soil Applications Affect Athelia rolfsii Stress Response in Sugar Beet (Beta vulgaris L.) Plants"

_plants, 2023, doi:10.3390/plants12193509_

Round 1
Reviewer 1 Report (Previous Reviewer 1)
This study estimated the “Effect of Zinc and Boron on Antioxidative Defense Mechanism in Sugar Beet (Beta Vulgaris L.) Plant under Athelia Rolfsii Stress”. This study do not contributes a new knowledge to plant physiology. Actually, there are many flaws in this MS that it is not suitable for publication in Plants in its current form.
I am disappointed with the lack of response to my requests for improving the manuscript. Requesting them once again seems like a waste of time to me. The authors did not address any of the serious issues of the paper that were pointed out to them. Unfortunately I find myself in the unpleasant position of insisting on the non-publication of the paper in its present form.
As I said before, Parameters which measured by authors is not enough to discuss their title.
If the authors want to get information about the behavior of the plant regarding to antioxidative defense mechanism they need also to measure the antioxidant enzymes [superoxide dismutase, guaiacol peroxidase, glutathione peroxidase, glutathione reductase, glutathione S-transferase, monodehydroascorbate reductase, dehydroascorbate] activity, non-enzymatic antioxidants [α-tocopherol, carotenoids, reduced glutathione, flavonoids, total phenols] content, ROS accumulation, protein oxidation, electrolyte leakage, membrane stability index, ...................................... to get a better inside into the influence of zinc and boron under Athelia Rolfsii stress on antioxidative defense mechanism.
The manuscript gives a few results. However it is far away to describe the mechanisms and the causal relationships. Authors should measure other parameters.

Moderate editing of English language required
Author Response
Please see the attachment.

Reviewer 2 Report (Previous Reviewer 2)
The manuscript has been significantly improved. I have no more comments.
Author Response
Please see the attachment

Reviewer 3 Report (Previous Reviewer 3)
This paper is improved as suggested and may published it with minor improvement in english.
Need to be minor english editing in respect of grammer.
Round 2
Reviewer 1 Report (Previous Reviewer 1)
The manuscript can be published.
Except: the Title should be changed to:
Zinc and Boron Soil Applications Affect Athelia Rolfsii Stress Response in Sugar Beet (Beta Vulgaris L.) Plants
Minor editing of English language required
Author Response
Please see the attachment

This manuscript is a resubmission of an earlier submission. The following is a list of the peer review reports and author responses from that submission.
Round 1
Reviewer 1 Report
This study estimated the “Effect of Zn and B on Antioxidative Defense Mechanism in Sugar Beet (Beta Vulgaris L.) Plant under Athelia Rolfsii Stress”. This study do not contributes a new knowledge to plant physiology.
Actually, there are many flaws in this MS that it is not suitable for publication in Plants in its current form. I have several major concerns with the manuscript which prevent me from recommending it for publication in its current situation.
· Parameters which measured by authors is not enough to discuss their title.
· If the authors want to get information about the behavior of the plant regarding to antioxidative defense mechanism they need also to measure the antioxidant enzymes [superoxide dismutase, guaiacol peroxidase, glutathione peroxidase, glutathione reductase, glutathione S-transferase, monodehydroascorbate reductase, dehydroascorbate] activity, non-enzymatic antioxidants [α-tocopherol, carotenoids, reduced glutathione, flavonoids, total phenols] content, ROS accumulation, protein oxidation, electrolyte leakage, membrane stability index, ...................................... to get a better inside into the influence of under Athelia Rolfsii stress on antioxidative defense mechanism.
· The manuscript gives a few results. However it is far away to describe the mechanisms and the causal relationships.
· Authors should measure other parameters, then they can resubmit the manuscript again.
Title:
(1) The MS title should be improved because “Antioxidative Defense Mechanism” is rather general and does not give information on the specific aspects of your work.
Indeed, The antioxidative defense mechanism is categorized into antioxidant enzymes [superoxide dismutase (SOD), guaiacol peroxidase (GPOX), catalase (CAT), ascorbate peroxidase (APX), glutathione peroxidase (GPX), glutathione reductase (GR), glutathione S-transferase (GST), monodehydroascorbate reductase (MDHAR), dehydroascorbate (DHAR)] and non-enzymatic antioxidants [α-tocopherol, carotenoids, ascorbic acid (AsA), reduced glutathione (GSH), flavonoids, total phenols].
However, authors only measured the antioxidant enzyme (CAT, POX and APX) activity and the content of ascorbic acid.
(2) Please do not use abbreviation (Zn and B) in the title.
Abstract:
(3) The basic experiment that was done by the authors is not clearly revealed in the abstract. I would advise the authors to re-write the abstract part focusing primarily on the foundation of the experiment they has undertaken.
(4) The presentation of the results should be carefully and completely revised.
(5) Language used in this section should be improved. There are a lot of linguistic mistakes.
Keywords:
(6) It in not enough, authors should also put Athelia Rolfsii Stress.
Introduction:
(7) There is not enough information on how sugar beet (Beta Vulgaris L.) respond to Athelia Rolfsii stress.
(8) Please illustrate by more details the impact of using Zn and B applications on the measured parameters under stressed and non-stressed conditions. Please incorporate recent references.
(9) At the end of this section, Please describe the novelty of your work in comparison with previous work. What has it added that we did not know before?
Material and methods:
(10) The aim of this study is to see the ‘Effect of Zn and B on Antioxidative Defense Mechanism in Sugar Beet (Beta Vulgaris L.) Plant under Athelia Rolfsii Stress’ as author mentioned in the manuscript title.
Parameters which measured by authors are not enough to discuss their title. To be honest I also expected to see data on the antioxidant enzymes [superoxide dismutase (SOD), guaiacol peroxidase (GPOX), glutathione peroxidase (GPX), glutathione reductase (GR), glutathione S-transferase (GST), monodehydroascorbate reductase (MDHAR), dehydroascorbate (DHAR)], non-enzymatic antioxidants [α-tocopherol, carotenoids, reduced glutathione (GSH), flavonoids, total phenols], reactive oxygen species (ROS) accumulation, protein oxidation, electrolyte leakage, membrane stability index, …………………… to get a better inside into the response of sugar beet (Beta Vulgaris L.) antioxidative defense mechanism under Athelia Rolfsii stress to Zn and B applications.
So, I am asking the authors to measured these parameters.
(11) How much plants were taken for different analysis should be mentioned clearly.
(12) In 2.5. Enzyme extractions and assays:
I am wondering under this title authors describe the methods of measuring the content of ascorbic acid and MDA ???!!!
Results:
(13) Results are poorly written.
Discussion:
(14) The discussion section is very descriptive and does not evaluate the biological significance of the results presented in any depth.
The authors make no attempt to explain the mechanism of action of Athelia Rolfsii stress as well as Zn and B applications. How Athelia Rolfsii stress as well as Zn and B applications affect the measured parameters. The actual mechanistic role of Zn and B under stressed and unstressed environments must be discussed with proper references. Many of the references are irrelevant.
(15) Throughout the manuscript, there is also a lack of indication of what is innovative in this paper and what the authors have contributed to the current state of knowledge.
Conclusion:
(16) The conclusion section is poorly written. Authors should include specific results of their research, which extend the current state of knowledge.
References
(17) References: need to be cross-checked.
Linguistic quality:
(18) The language quality is so poor and this paper must be edited by professional English editor.

Extensive editing of English language required
Reviewer 2 Report
The manuscript deals with the assessment of the role of zinc and boron in the tolerance of sugar beet to Athelia rolfsii infection. Some references should be replaced by newer ones. The details are listed below:
L23: italics for all Latin names throughout the paper
L24-30: add some % changes of examined parameters between treatments
L51: indicate subscripts and check throughout (O2, H2O2 etc.)
L110-117: whether the plants were cultivated in phytotron? Indicate photoperiod, temperature, volume of water used to irrigation and how often the plants were irrigated?
L147: volume of sample
L151: volume of enzyme extract
L154: volume of plant sample, acetic acid, volume and concentration of meta phosphate
L157: oxidative damage
L201-202: why the Authors compare inoculated plants to uninoculated control? The effect of Zn or B should be compared within the group of inoculated treatments to inoculated control and in opposite way. Similar comparisons are included in the entire Results section.
L291-292: rephrase
L307-308: rephrase
L310-311: MDA content decreased in inoculated plants as shown in Fig. 7
L326: include newer reference instead 39: https://doi.org/10.1007/s00425-022-03838-x
L328-329: …micronutrients and organic compounds suppress… Add a newer reference instead 45: https://doi.org/10.3390/agronomy13051378
L336: A. rolfsii
L380: physical barrier
L413: some other mechanisms – indicate them
Extensive editing of English language required
Reviewer 3 Report
Line 23.. ‘A. rolfsii’ should be italicized.
Line 36.. Italicizes the text ‘(Beta vulgaris)’
Line 37.. ‘This crop covers almost 30% for global human consumptive sugars’ rewrite the sentence.
Line 51.. (1O2), H2O2,…use subscript
Line 53… write ‘able to damage the cell’ instead of ‘able to damage to the cells’
Line 349..incorporate causes instead of reasons and rewrite the sentence.
Line 357.. ‘Tufail et al’ italicized et al.
Line 360 - 363.. rewrite the sentence.
Line 412.. infection differently?
minor corrections in respect of spellings and some places in grammer